# Beyond Joint Demonstrations: Personalized Expert Guidance for Efficient Multi-Agent Reinforcement Learning

**Peihong Yu**                                                                *peihong@umd.edu*
*University of Maryland*

**Manav Mishra**                                                             *mishra20@iiserb.ac.in*
*IISER Bhopal*

**Alec Koppel**                                                          *alec.koppel@jpmchase.com*
*JP Morgan Chase & Co.*

**Carl Busart**                                                      *carl.e.busart.civ@army.mil*
*DEVCOM Army Research Laboratory*

**Priya Narayan**                                                 *priya.narayanan.civ@army.mil*
*DEVCOM Army Research Laboratory*

**Dinesh Manocha**                                                          *dmanocha@umd.edu*
*University of Maryland*

**Amrit Bedi**                                                               *amritbedi@ucf.edu*
*University of Central Florida*

**Pratap Tokekar**                                                           *tokekar@umd.edu*
*University of Maryland*

**Reviewed on OpenReview:** *https://openreview.net/forum?id=kzPNHQ8ByY*

## Abstract

Multi-Agent Reinforcement Learning (MARL) algorithms face the challenge of efficient exploration due to the exponential increase in the size of the joint state-action space. While demonstration-guided learning has proven beneficial in single-agent settings, its direct applicability to MARL is hindered by the practical difficulty of obtaining *joint* expert demonstrations. In this work, we introduce a novel concept of *personalized expert demonstrations*, tailored for each individual agent or, more broadly, each individual *type* of agent within a heterogeneous team. These demonstrations solely pertain to single-agent behaviors and how each agent can achieve personal goals without encompassing any cooperative elements, thus naively imitating them will not achieve cooperation due to potential conflicts. To this end, we propose an approach that *selectively* utilizes personalized expert demonstrations as guidance and allows agents to learn to cooperate, namely personalized expert-guided MARL (PegMARL). This algorithm utilizes two discriminators: the first provides incentives based on the alignment of individual agent behavior with demonstrations, and the second regulates incentives based on whether the behaviors lead to the desired outcome. We evaluate PegMARL using personalized demonstrations in both discrete and continuous environments. The experimental results demonstrate that PegMARL outperforms state-of-the-art MARL algorithms in solving coordinated tasks, achieving strong performance even when provided with suboptimal personalized demonstrations. We also showcase PegMARL's capability of leveraging joint demonstrations in the StarCraft scenario and converging effectively even with demonstrations from non-co-trained policies.

# 1  Introduction

The use of expert demonstrations[1] has been proven effective in accelerating learning in single-agent reinforcement learning, as evidenced by studies such as Kang et al. (2018); Chen & Xu (2022); Rengarajan et al. (2022). This approach has since been extended to Multi-Agent Reinforcement Learning (MARL) (Lee & Lee, 2019; Qiu et al., 2022), which typically assumes the availability of high-quality collaborative **joint demonstrations**. However, from a practical standpoint, collecting joint demonstrations can be labor-intensive, demanding one user per agent in cooperative scenarios. Furthermore, these demonstrations are not scalable. If we change the *number* of agents or introduce new *types* of agents, we will need to gather a new set of demonstrations to learn from.

In contrast, it is much easier to obtain demonstrations for individual agents, or even better, for each *type* of agents in a heterogeneous setting. We thus ask the following research question: *could we leverage individual-wise task demonstrations instead?* In this work, we refer to such expert demonstrations that address single-agent behaviors for personal objectives as **personalized demonstrations** (see Figure 1). Since the personalized demonstrations will not necessarily reflect how the agents can collaborate and may even conflict with each other in the joint setting, naively mimicking the demonstrations will not achieve cooperation. Therefore, purely imitation learning-based approaches would not be effective. We need an approach that *selectively* utilizes suitable personalized expert demonstrations as guidance and allows agents to learn to cooperate via collecting reward signals from the environments.

To this end, we present our algorithm, Personalized Expert-Guided MARL (PegMARL), which carries out personalized occupancy matching as a form of guidance through reward-shaping. We implement this via two discriminators. The first, a personalized behavior discriminator, evaluates local state-action pairs, providing positive incentives for actions that align with the demonstration and negative incentives for divergent ones. The second, a personalized transition discriminator, assesses whether a local state-action pair induces a desired change in dynamics similar to that observed in the demonstration, adjusting the incentive weight accordingly. We demonstrate the effectiveness of PegMARL on both discrete gridworld and continuous multi-agent particle environments (Lowe et al., 2017; Mordatch & Abbeel, 2017). The main contributions of this paper are summarized as follows:

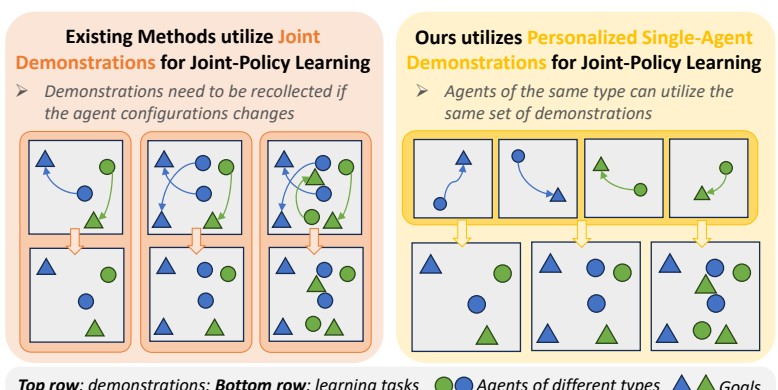

Figure 1: Joint demonstrations are costly to collect but offer rich information on collaborative behaviors. Personalized demonstrations are easier to collect, but solely focus on individual agent goals, so they lack cooperative elements.

(1) We propose PegMARL, the first approach that enables utilizing **personalized demonstrations** for policy learning in heterogeneous MARL environments, which (i) avoids demonstration recollection regardless of the *number* and *type* of agents involved, and (ii) is compatible with most MARL policy gradient methods.

(2) We demonstrate PegMARL's effectiveness with **personalized demonstrations** in both **discrete** and **continuous** environments. Our algorithm outperforms state-of-the-art decentralized MARL algorithms, pure multi-agent imitation learning, and reward-shaping techniques in terms of scalability and convergence speed, and achieves robust performance even when provided with suboptimal demonstrations.

---

[1]In our context, the term "expert guidance" refers to demonstrations that provide meaningful, above-random performance in complex environments, even if the demonstrations are not necessarily optimal.

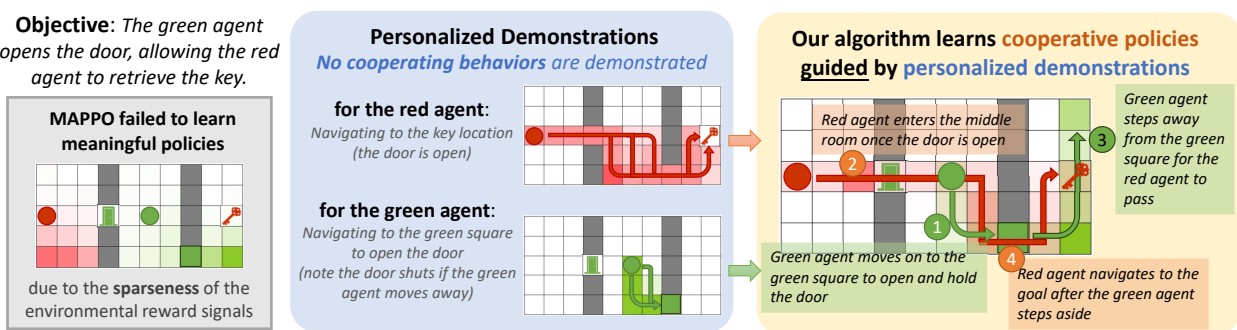

Figure 2: An example of utilizing **personalized** demonstrations to learn **cooperative** multi-agent policies. To learn successful cooperation, the agents are required not only to imitate the demonstrations to achieve personal goals but also to learn how to avoid conflicts and collaborate. We visualize the state visitation frequency of the personalized demonstrations and the joint policies learned by our algorithm and MAPPO, where a darker color means a higher value. We observe that the demonstrations guide the agents in exploring the state space more efficiently than in MAPPO.

(3) We showcase PegMARL's capability to also leverage **joint demonstrations**, regardless of whether they are sampled from co-trained or non-co-trained policies[2]. Experimental results on the StarCraft environment (Samvelyan et al., 2019) demonstrate that PegMARL converges effectively even with demonstrations from non-co-trained policies, which include potentially conflicting behaviors.

## 2 Related Works

**Imitation Learning (IL).** IL methods seek to replicate an expert policy from demonstration data generated by that policy. Behavior Cloning (BC) is a commonly employed IL technique (Pomerleau, 1991; Bojarski et al., 2016), where the expert policy is estimated through supervised learning on demonstration data. However, BC is susceptible to compounding errors resulting from distribution shifts (Ross et al., 2011). Another thread of IL research is Inverse Reinforcement Learning (IRL) (Ng et al., 2000; Ziebart et al., 2008), in which the underlying reward function is estimated from the demonstration data and then used for policy learning. To alleviate the computational overhead of IRL, Generative Adversarial Imitation Learning (GAIL) (Ho & Ermon, 2016) was introduced, allowing direct policy learning from observed data without intermediate IRL steps. Song et al. (2018) extended this approach to introduce Multi-Agent GAIL (MAGAIL), which adapts GAIL to high-dimensional environments featuring multiple cooperative or competing agents. In general, IL approaches can rarely perform better than demonstrations. Therefore, they are not directly suitable for scenarios where only personalized expert demonstrations that do not demonstrate how to collaborate are available.

**Learning from Demonstration (LfD).** In contrast to IL methods, LfD aims to leverage demonstration data to facilitate learning rather than simply mimicking expert behavior. Existing LfD works incorporate demonstration data into a replay buffer with a prioritized replay mechanism to accelerate the learning process. Due to the off-policy nature of the demonstration data, most methods are value-based (Hester et al., 2018; Vecerik et al., 2017). There has been some recent work where the demonstrations are used to aid exploration, especially in environments with large state-action spaces (Kang et al., 2018; Chen & Xu, 2022; Rengarajan et al., 2022). For instance, POfD (Kang et al., 2018) learns an implicit reward from the demonstration data using a discriminator and incorporates it into the original sparse reward. LOGO (Rengarajan et al., 2022) uses the demonstration data to directly guide the policy update: during each update iteration, the algorithm seeks a policy that closely resembles the behavior policy within a trust region. However, these approaches are primarily focused on single-agent settings.

---

[2]"Co-trained policies" refers to policies that have been trained together in the same environment with shared experiences. We refer the readers to Figure 3 for an illustration and Section 6.2 of Wang et al. (2023) for more information.

In multi-agent settings, Qiu et al. (2022) suggest using demonstrations to pretrain agents through imitation learning as a warm start, followed by optimization of the pretrained policies using standard MARL algorithms. Lee & Lee (2019) augment the experience buffer with demonstration trajectories and gradually decrease the mixing of demonstration samples during training to prevent the learned policy from being overly influenced by demonstrations. Similar to POfD, $DM^2$ (Wang et al., 2023) enables agents to enhance their task-specific rewards by training discriminators as well. Each agent matches toward a target distribution of concurrently sampled trajectories from a joint expert policy to facilitate coordination. These approaches, however, require joint demonstrations of the same team configurations sampled from cohesively trained policies. While effective, this requirement can be cumbersome and limiting, especially in real-world scenarios where such coordinated demonstrations are difficult to obtain (as mentioned in Section 1).

Our approach differs in that we leverage only personalized expert demonstrations to learn a cooperative policy. Notably, PegMARL can also utilize joint demonstrations when they are available, including those from non-co-trained policies. By accommodating these diverse and potentially conflicting demonstrations, PegMARL can leverage a wider range of available data, offering greater flexibility and robustness across various multi-agent applications.

## 3 Preliminaries

We start by considering a Markov Decision Process $(\mathcal{S}, \mathcal{A}, P, r, \gamma)$ for a cooperative multi-agent setting with $N$ agents. Here, $\mathcal{S}$ denotes the global state across all the agents, which can be decomposed as the product of $N$ local spaces $\mathcal{S}_i$ as $\mathcal{S} = \mathcal{S}_1 \times \mathcal{S}_2 \times \cdots \times \mathcal{S}_N$. By noting the local state of agent $i$ as $s_i \in \mathcal{S}_i$, the global state is $s = (s_1, s_2, \cdots, s_N)$. Similarly, we define the global action space $\mathcal{A}$ as $\mathcal{A} = \mathcal{A}_1 \times \mathcal{A}_2 \times \cdots \times \mathcal{A}_N$, meaning that for any $a \in \mathcal{A}$, we may write $a = (a_1, a_2, \cdots, a_N)$ with $a_i \in \mathcal{A}_i$. The transition probability from state $s$ to $s'$ after taking a joint action $a$ is denoted by $P(s'|s,a) = \prod_i P_i(s'_i|s,a)$. All agents share a common reward function $r = R(s,a)$, and $\gamma \in [0,1]$ is the discount factor. In this work, we focus on decentralized learning and define the global policy as $\pi_\theta(a|s)$, where $\theta \in \Theta$ are the policy parameters. Specifically, we have $\theta = (\theta_1, \theta_2, \cdots, \theta_N)$ as the factorized global policy parameters, and we can write $\pi_\theta(a|s) = \prod_i \pi_{\theta_i}(a_i|s)$ using policy factorization.

**Personalized Tasks and MDPs.** To introduce the notion of **personalized demonstration**, we need to extract the individual tasks of each agent from the collective tasks of multiple agents. For example, in Figure 2, the green agent's personalized task is to open the door and the red agent's personalized task is to reach the key *without the other's presence*. We then define Personalized Markov Decision Processes (PerMDPs) $(\mathcal{S}_i, \mathcal{A}_i, Q_i, r_i, \gamma)$ for each agent or, more generally, for each *type* of agents that share the same objective within a heterogeneous team. Here, a *type* refers to behaviorally identical agents who share the same state and action spaces, as well as the same reward function, using the formalism from Bettini et al. (2023), and we provide the PerMDP definition for each agent for simplicity. We assume that the state space $\mathcal{S}_i$ and action space $\mathcal{A}_i$ of the personalized task $\mathcal{T}_i$ are the same as the local state and action spaces of agent $i$ from the joint task. The transition probability from state $s_i$ to $s'_i$ after taking a action $a_i$ is represented as $Q_i(s'_i|s_i, a_i)$. By following an arbitrary policy $\pi_i(a_i|s_i)$ for this personalized task, we can collect a set of personalized demonstrations $\mathcal{B}_i = \{(s_i^t, a_i^t)\}_{t=0}^H$, where $H$ is the episode horizon. We'd like to emphasize that while we assume the personalized tasks and the joint task are conducted in the same environment map, the underlying transition dynamics are different. Additionally, the joint reward $r$ may not necessarily equal the summation of rewards $r_i$ for each personalized task.

**Occupancy Measures.** For a joint policy $\pi = (\pi_1, \pi_2, ..., \pi_N)$, we can write the global state action occupancy measure $\lambda^\pi(s,a)$ as

$$\lambda^\pi(s,a) = \sum_{t=0}^\infty \gamma^t \cdot \mathbb{P}\Big(s^t = s, a^t = a \mid \pi\Big) \tag{1}$$

and write the corresponding local cumulative state-action occupancy measure as

$$\lambda_i^\pi(s_i, a_i) = \sum_{t=0}^\infty \gamma^t \cdot \mathbb{P}\Big(s_i^t = s_i, a_i^t = a_i \mid \pi\Big) \tag{2}$$

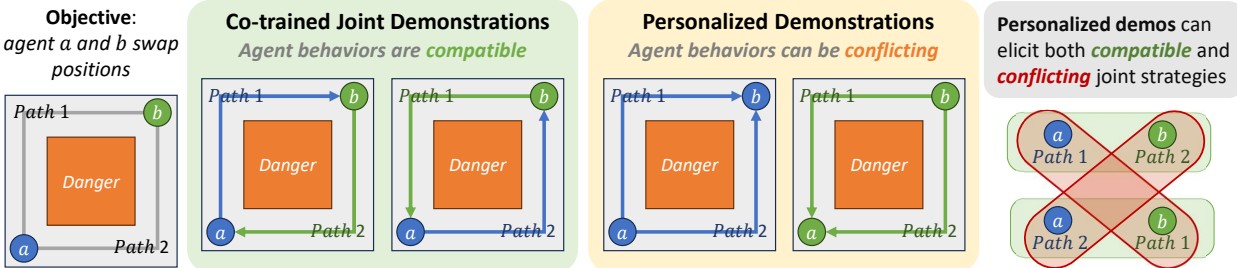

Figure 3: When joint demonstrations are sampled from co-trained policies, the agents' behaviors exhibit compatibility. In contrast, personalized demonstrations solely focus on how each agent achieves its individual goal and lack cooperative elements, potentially leading to conflicts.

for $\forall a_i \in \mathcal{A}_i, s_i \in \mathcal{S}_i$. An interesting observation to note here is that we can write the local occupancy measure as the marginalization of the global occupancy measure with respect to all other agents. Mathematically, it holds that

$$\lambda_i^\pi(s_i, a_i) = \sum_{a \in \{a_i\} \times \mathcal{A}_{-i}} \sum_{s \in \{s_i\} \times \mathcal{S}_{-i}} \lambda^\pi(s, a) \tag{3}$$

with $\mathcal{A}_{-i} = \Pi_{j \neq i} \mathcal{A}_j$ and $\mathcal{S}_{-i} = \Pi_{j \neq i} \mathcal{S}_j$.

## 4 MARL with Personalized Expert Demonstrations

Now, we are ready to present the main problem we are interested in solving in this work. Assume each agent $i$ is associated with a personalized task $\mathcal{T}_i$. We collect one set of expert demonstrations for each agent $i$ or, equivalently, each personalized task $\mathcal{T}_i$. By letting an expert user perform each personalized task in the respective personalized MDP, we obtain a collection of expert demonstrations denoted by $\{\mathcal{B}_{E_1}, \mathcal{B}_{E_2}, ..., \mathcal{B}_{E_N}\}$. We assume that the underlying expert policy associated with $\mathcal{B}_{E_i}$ is $\pi_{E_i}(a_i|s_i)$, and $\lambda^{\pi_{E_i}}$ is the occupancy measure following the expert's policy $\pi_{E_i}$ for agent $i$. Note that while we establish the problem formulation and develop our algorithm in a fully observable setting, the experiments are conducted in both fully and partially observable settings, encompassing both discrete and continuous environments (details in Section 5).

### 4.1 Formulating Personalized Expert-Guided MARL

In standard Multi-Agent Reinforcement Learning, agents aim to discover optimal joint policies that maximize the long-term return $R(\pi_\theta) := \langle \lambda^{\pi_\theta}, r \rangle = \frac{1}{N} \sum_{i=1}^N \langle \lambda_i^{\pi_\theta}, r \rangle$. Typically, the learning process commences with random exploration, which is often inefficient due to MARL's exponentially growing exploration spaces, especially when rewards are sparse. Beginning with the intuition of leveraging personalized demonstrations as guidance for how each agent should accomplish their personalized tasks to promote more effective exploration, akin to Kang et al. (2018) for single-agent cases, we can define the objective for learning from personalized demonstrations in multi-agent settings as follows:

$$\max_{\theta \in \Theta} \frac{1}{N} \sum_{i=1}^N \left( \langle \lambda_i^{\pi_\theta}, r \rangle - \eta \mathbb{D}_{\mathrm{JS}} \left( \lambda_i^{\pi_\theta} \ || \ \lambda^{\pi_{E_i}} \right) \right), \tag{4}$$

where $\eta$ is a weighting term balancing the long-term reward and the personalized policy similarity. The Jensen-Shannon (JS) Divergence terms enable individual agents to align their actions with their respective personalized demonstrations and facilitate the achievement of their specific objectives.

However, only imitating the personalized demonstration may not always yield favorable outcomes and can even impede the learning process. Previous works in MARL have predominantly utilized joint demonstrations as guidance. As demonstrated by DM$^2$ (Wang et al. (2023)), trajectories must be sampled from a *co-trained* joint expert policy for their joint action-matching objective to converge. The crucial aspect that jointly coordinated

demonstrations offer, which personalized demonstrations lack, is compatibility. To illustrate, consider the example depicted in Figure 3, where two agents aim to swap positions without entering the danger region. Two sets of optimal joint policies exist for this task: agent $a$ takes path 1 and agent $b$ takes path 2, or vice versa.

When sampling joint demonstrations from a set of co-trained policies, the agents' behaviors will be naturally compatible. In contrast, if personalized demonstrations were provided, agents could indifferently choose between both paths to reach their goals, potentially resulting in conflicting strategies through naive imitation.

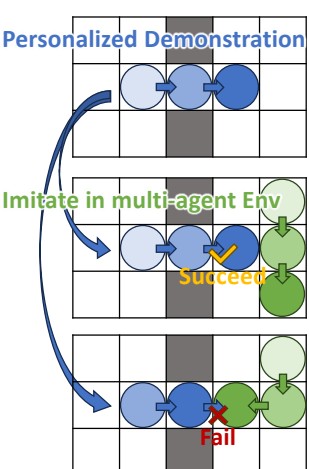

Hence, we pose the following question: *How can we leverage the benefits of using personalized demonstrations to enhance exploration efficiency while circumventing the imitation of conflicting behaviors that might hinder joint-policy training?*

Consider the scenario depicted in Figure 4, where the blue agent seeks to replicate its personalized demonstration within a multi-agent environment. The blue agent can successfully transition to the neighboring grid on the right if it's unoccupied; otherwise, the transition fails. Viewing the blue agent as the focal agent and the green agent as part of the environment, successful imitation hinges on the agent's "local transition" matching those of the environment where the personalized demonstrations originated. We define $P(s_i'|s_i, a_i)$ as an approximation for the $i$-th agent's *local transition probability* in the multi-agent environment, which is governed by the true transition dynamics $P(s'|s, a)$ and the policy $\pi_\theta$. With this insight, we further refine the objective function as follows:

Figure 4: A motivating example illustrating the imitation of personalized demonstrations in a multi-agent environment. The primary technical challenge lies in the discrepancy between the transition dynamics in the personalized MDP and the local transition dynamics for each agent in the multi-agent environment.

$$\max_{\theta \in \Theta} \frac{1}{N} \sum_{i=1}^{N} \left( \langle \lambda_i^{\pi_\theta}, r \rangle - \eta \mathbb{D}_{\mathrm{JS}} \left( \hat{\lambda}_i^{\pi_\theta} \ || \ \hat{\lambda}^{\pi_{E_i}} \right) \right), \qquad (5)$$

where $\hat{\lambda}_i^{\pi_\theta}$ and $\hat{\lambda}^{\pi_{E_i}}$ have the same definition as the original $\lambda_i^{\pi_\theta}$ and $\lambda^{\pi_{E_i}}$ but with a restricted domain:

$$\mathrm{dom}_{\hat{\lambda}_i^{\pi_\theta}} = \mathrm{dom}_{\hat{\lambda}_i^{\pi_{E_i}}} = \{(s_i, a_i) \in \mathcal{S}_i \times \mathcal{A}_i | \mathbb{D}_{JS} \left( P_i(s_i'|s_i, a_i) \ || \ Q_i(s_i'|s_i, a_i) \right) \leq \epsilon\}. \qquad (6)$$

where $Q(s_i'|s_i, a_i)$ represents the true transition dynamics of the personalized MDP from which demonstrations are collected, and $\epsilon$ is a threshold parameter that theoretically controls transition mismatch tolerance. By adjusting the learning objective in this manner, the occupancy matching only happens on those local state-action pairs that guide us toward the desired next local state under the current policy.

## 4.2 Solving Personalized Expert-Guided MARL

Now, we can begin the development of a practical algorithm. By adopting Theorem 2 from Kang et al. (2018)[3], we can substitute the JS divergence between occupancy measures with

$$\mathbb{D}_{\mathrm{JS}} \left( \lambda_i^{\pi_\theta} \ || \ \lambda^{\pi_{E_i}} \right) \approx \sup_{D_i} \left( \mathbb{E}_{\pi_\theta}[\log(1 - D_i(s_i, a_i))] + \mathbb{E}_{\pi_{E_i}}[\log(D_i(s_i, a_i))] \right), \qquad (7)$$

where $D(s_i, a_i) : \mathcal{S}_i \times \mathcal{A}_i \to (0, 1)$ is a discriminative classifier to discern if the $(s_i, a_i)$ pair is from the demonstration or the current policy. However, this doesn't account for the constraint in equation 6. Therefore, by introducing an indicator function $\mathbb{1}(s_i, a_i)$ for whether $(s_i, a_i)$ belongs to the domain of $\hat{\lambda}_i^{\pi_\theta}$ and $\hat{\lambda}^{\pi_{E_i}}$, we can obtain

$$\mathbb{D}_{\mathrm{JS}} \left( \hat{\lambda}_i^{\pi_\theta} \ || \ \hat{\lambda}^{\pi_{E_i}} \right) \approx \sup_{D_i} \left( \mathbb{E}_{\pi_\theta} \left[ \mathbb{1}(s_i, a_i) \cdot \log(1 - D_i(s_i, a_i)) \right] + \mathbb{E}_{\pi_{E_i}} \left[ \mathbb{1}(s_i, a_i) \cdot \log(D_i(s_i, a_i)) \right] \right). \qquad (8)$$

---

[3]We refer the reader to Kang et al. (2018) for the full derivation. We use the same calculations with the change of notation $\rho_\pi \leftarrow \lambda_i^{\pi_\theta}$ and $\rho_E \leftarrow \lambda^{\pi_{E_i}}$.

---

**Algorithm 1 P**ersonalized **E**xpert-**G**uided MARL (PegMARL)

---

**Input:** Number of agents $N$; environment $env$; personalized expert trajectories $\mathcal{B}_{E_1}, ..., \mathcal{B}_{E_N}$; batch size $M$; weight parameters $\{\eta^k\}$.
**Initialize:** Policies $\{\pi_{\theta_i}\}$, discriminators $\{D_{\phi_i}\}$ and $\{D_{\bar{\phi}_i}\}$, where $i = 1, 2, ..., N$.
**Output:** Learned policies $\{\pi_{\theta_i}\}$.

1: **for** iteration $k = 0, 1, 2, ...$ **do**
2:     Gather trajectories of multi-agent rollouts from $env$, $\mathcal{B}^k = ROLLOUT(\pi, env)$.
3:     **for** agent $i = 0, 1, ..., N - 1$ **do**
4:         Sample $M$ tuples of $(s_i, a_i, s'_i)$ from demonstration $\mathcal{B}^{E_i}$ and $\mathcal{B}^k$;
5:         Update personalized behavior discriminator $D_{\phi_i}$:

$$\max_{\phi_i} \left( \mathbb{E}_{\mathcal{B}^k} \left[ \log(1 - D_{\phi_i}(s_i, a_i)) \right] + \mathbb{E}_{\mathcal{B}_{E_i}} \left[ \log D_{\phi_i}(s_i, a_i) \right] \right). \tag{9}$$

6:         Update personalized transition discriminator $D_{\bar{\phi}_i}$:

$$\max_{\bar{\phi}_i} \left( \mathbb{E}_{\mathcal{B}^k} \left[ \log(1 - D_{\bar{\phi}_i}(s_i, a_i, s'_i)) \right] + \mathbb{E}_{\mathcal{B}_{E_i}} \left[ \log D_{\bar{\phi}_i}(s_i, a_i, s'_i) \right] \right). \tag{10}$$

7:         Estimate the reshaped reward as $\hat{r}_i^k = r - \eta^k D_{\bar{\phi}_i}(s_i, a_i, s'_i) \log(1 - D_{\phi_i}(s_i, a_i))$.
8:         Update agent policy $\pi_{\theta_i}$.
9:     **end for**
10: **end for**

---

Since direct access to the transition distributions is unavailable, verifying whether each $(s_i, a_i)$ pair is within the domain is not directly feasible. To address this, we further approximate the indicator function using another discriminative classifier $D_i(s_i, a_i, s'_i) : \mathcal{S}_i \times \mathcal{A}_i \times \mathcal{S}_i \to (0, 1)$, estimating the likelihood of a $(s_i, a_i, s'_i)$ tuple being from the demonstrations. In essence, it quantifies the likelihood that the corresponding $(s_i, a_i)$ induces a desired change in local state transition as observed in the demonstrations, effectively serving as a learned proxy for the $\epsilon$-constrained domain in Equation 6.

After approximating the inner JS divergence term in equation 5 with two discriminators, we integrate the discriminative rewards derived from these discriminators into the environmental reward for the outer problem. Specifically, we estimate the reshaped reward $\hat{r}_i$ as $\hat{r}_i = r - \eta D_{\bar{\phi}_i}(s_i, a_i, s'_i) \log(1 - D_{\phi_i}(s_i, a_i))$, where $\phi_i$ and $\bar{\phi}_i$ are parameters for the two discriminators. The personalized *behavior* discriminator $D_{\phi_i}$ evaluates local state-action pairs, providing positive incentives for actions that align with the demonstration and negative incentives for divergent ones, while the personalized *transition* discriminator $D_{\bar{\phi}_i}$ assesses if a local state-action pair induces a desired transition in local state akin to that observed in the demonstration, adjusting the incentive weight accordingly. Subsequently, policy optimization is conducted by maximizing the long-term return with the reshaped reward.

Our PegMARL algorithm, detailed in Algorithm 1, is compatible with any policy gradient methods, with MAPPO (Yu et al. (2021)) adopted in our implementation. While presented in a fully observable setting, PegMARL can be adapted for partially observable scenarios. In such cases, we process observations from joint environment rollouts by removing dimensions not observable in the Personalized MDP for discriminators inputs. Notably, PegMARL can also utilize joint demonstrations; in this case, the observation processing step is unnecessary as both demonstrations and environment rollouts are derived from the same joint MDP. Compared to DM$^2$ (Wang et al., 2023), which also uses a discriminator-based approach for multi-agent learning from demonstrations, PegMARL introduces the personalized transition discriminator $D_{\bar{\phi}_i}$ as a key innovation. This additional discriminator helps address the challenge illustrated in Figure 3 by evaluating whether local state-action pairs induce desired transitions similar to those in the demonstrations, effectively filtering out less desirable actions while retaining beneficial ones. This component enables PegMARL to handle both personalized demonstrations and joint demonstrations from various sources, including those

sampled from non-co-trained policies, while DM$^2$ requires demonstrations from co-trained policies to achieve convergence.

## 5 Experiments

In this section, we empirically evaluate the performance of PegMARL, focusing on the following questions: (1) How does PegMARL, which leverages personalized demonstrations, compare to state-of-the-art MARL techniques? (2) How does PegMARL scale with an increasing number of agents and in the case of continuous state-action spaces? (3) How does the sub-optimality of personalized expert demonstrations affect the performance of PegMARL? and (4) How does PegMARL perform when trained with joint demonstrations sampled from co-trained or non-co-trained expert policies?

### 5.1 Main Results with Personalized Demonstrations

We evaluate PegMARL with personalized demonstrations on discrete gridworld environments (Figure 5) and a continuous multi-agent particle environment (MPE, Figure 8a). The gridworld environment is fully observable with discrete state and action spaces, while the MPE is partially observable with a continuous state space and discrete action space. Both environments feature sparse reward signals, which adds to the challenge for standard MARL algorithms to learn effectively.

Our comparison includes four strong baselines: MAPPO (Yu et al. (2021)), a leading decentralized MARL algorithm; MAGAIL (Song et al. (2018)), a state-of-the-art multi-agent imitation learning algorithm; DM$^2$ (Wang et al. (2023)), which combines distribution matching reward with environmental reward; and ATA (She et al. (2022)), one of the best multi-agent reward-shaping methods. Notably, MAGAIL and DM$^2$ were not originally designed for personalized demonstrations. Hence, we have adapted them for use in personalized demonstration settings through necessary modifications. Specifically, we processed the observations from joint environment rollouts by removing dimensions not observable in the Personalized MDP, similar to our approach in PegMARL. This adjustment ensures that the inputs to their respective discriminators are compatible with the personalized demonstration data. In this context, MAGAIL serves as an ablation of PegMARL lacking the environmental signal and transition discriminator, while DM$^2$ represents an ablation lacking only the transition discriminator. Details on algorithm implementation and hyperparameter choices can be found in Appendix C.

For each scenario, we collect two sets of personalized demonstrations for each agent: *optimal* and *suboptimal*, by training single-agent RL in simplified environments that focus on each agent's specific tasks. The average episodic rewards of suboptimal demonstrations are approximately half of their optimal counterparts (more details can be found in Appendix B). We execute each method in every environment with 10 distinct random initializations and plot the mean and variance across all runs. Oracle denotes the best possible return achievable with optimal policies.

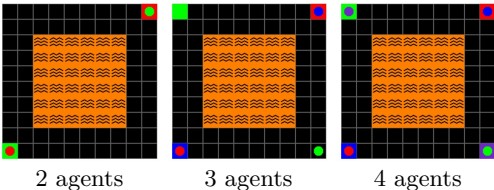

(a) **The lava scenario**: the agents are homogeneous, aiming to reach corresponding diagonal positions without entering the lava. The episode ends if any agent steps into the lava.

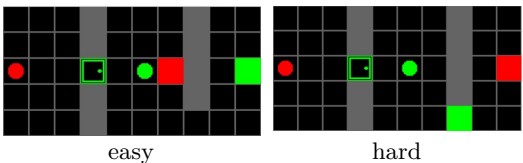

(b) **The door scenario**: the agents are heterogeneous, the assistant agent (green) must reach the green square and remain there to hold the door open, allowing the other agent (red) to reach the goal.

Figure 5: The discrete gridworld environments. Circles indicate the starting locations of the agents, while squares of the same color denote their respective goal locations. More details can be found in Appendix A.1.

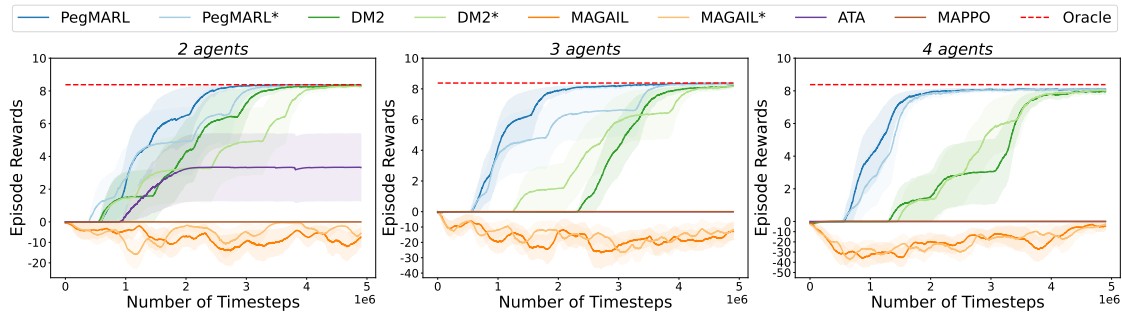

Figure 6: Learning curves of PegMARL versus other baseline methods under the **lava scenario**. PegMARL converges to higher rewards and generalizes better to larger numbers of agents. The star symbols (*) in the legend indicate that suboptimal personalized demonstrations are adopted.

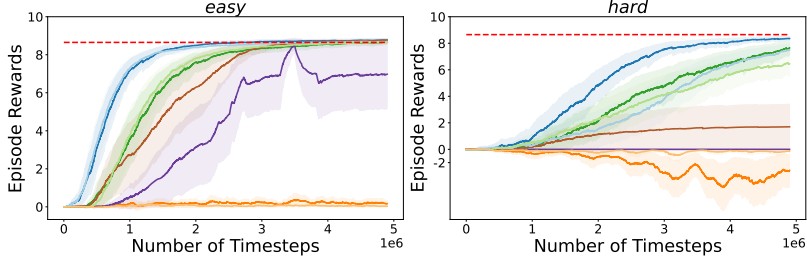

Figure 7: Learning curves of PegMARL versus other baseline methods under the **door scenario**. PegMARL shows better robustness in terms of convergence and generalizability.

**How does PegMARL scale with an increasing number of agents?** The lava scenario (Figure 5a) contains three variations, each involving different quantities of agents and escalating levels of complexity. The agents receive a positive reward only upon reaching the goal, incur penalties for collisions, and instantly die and terminate the episodes if they step into the lava, making learning in this environment challenging. As we observe from Figure 6, MAPPO struggles to develop meaningful behavior across all scenarios due to the sparse reward structure. MAGAIL performs worse, primarily due to the absence of environmental reward signals. The naive imitation of personalized agent demonstrations leads to frequent collisions among agents. While ATA can learn suboptimal policies in the 2-agent setting, its learning efficacy declines drastically as the number of agents increases. This underscores the challenges of training as the complexity of the multi-agent environment increases.

In contrast, PegMARL demonstrates superior generalizability across scenarios with more agents, maintaining stable performance despite suboptimal demonstrations. As DM$^2$ has been adapted to accommodate personalized demonstrations, it closely resembles PegMARL but lacks the personalized transition discriminator component. As the number of agents increases, we would expect the space of possible interactions among agents to expand, and therefore for the personal demonstrations to be potentially misleading. Consequently, DM$^2$ experiences a decrease in convergence speed as the number of agents increases. This highlights the importance of the personalized transition discriminator in PegMARL, enabling effective handling of inter-agent interactions in personalized demonstration scenarios, thereby ensuring its efficacy across varying agent counts.

**How does PegMARL perform under the heterogeneous setting?** The door scenario (Figure 5b) contains two variants with varying levels of difficulty. The easy case is fairly straightforward: success should be achievable by adhering to personalized demonstrations, which illustrate how each agent navigates to their respective goal locations. As shown in Figure 7, most algorithms, except for MAGAIL, showcase proficient performance. Notably, PegMARL exhibits the swiftest convergence. We attribute MAGAIL's failure in this case to its inability to direct the green agent to remain positioned at the green square — a behavior not explicitly demonstrated. This underscores the importance of environmental reward signals when integrating personalized demonstrations into multi-agent learning paradigms. The hard case necessitates a higher degree

of agent cooperation: once the red agent gains entry to the middle room, the green agent must move aside from the green square to enable the red agent's passage into the right room. In this complex setting, only PegMARL and DM² demonstrate commendable convergence. PegMARL, equipped with the personalized transition discriminator, maintains faster convergence compared to DM².

**How does PegMARL perform in the continuous setting?** We modified the cooperative navigation task from the multi-agent particle environment (Lowe et al. (2017); Mordatch & Abbeel (2017)) to evaluate the performance of our algorithm in a continuous environment (more details can be found in Appendix A.2). Figure 8b demonstrates the learning curves of PegMARL versus MAPPO and DM². This validates the ability of PegMARL in continuous environments.

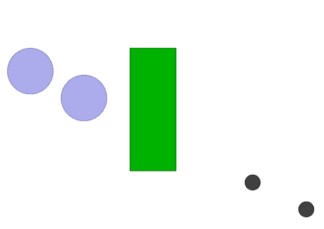

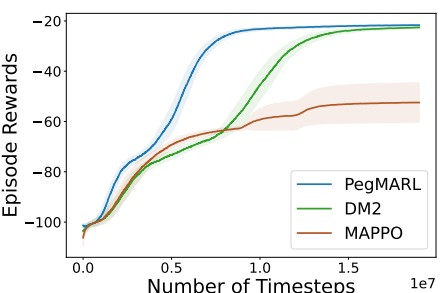

(a) The modified cooperative navigation scenario: the agents need to navigate around the wall to occupy both landmarks.

(b) Learning curves of PegMARL versus MAPPO and DM² under the modified cooperative navigation scenario

Figure 8: (a) The modified cooperative navigation scenario. (b) The learning curves demonstrate that PegMARL is effective in continuous environments.

## 5.2 Results with Joint Demonstrations

While PegMARL is primarily designed to leverage personalized demonstrations, it can also be extended to utilize joint demonstrations. Unlike other MARL algorithms like DM² (Wang et al. (2023)) that require compatible joint demonstrations sampled from co-trained policies to achieve convergence, PegMARL has the flexibility to utilize joint demonstrations even from non-co-trained policies, which potentially contain conflicting behaviors. To demonstrate, we conduct comparative experiments within the StarCraft Multi-Agent Challenge (SMAC) environment (Samvelyan et al. (2019)). We adopted DM²'s original implementation and parameters for accurate results replication. We employed the same two tasks that were used in DM²: 5mv6m, which involves 5 Marines (allies) against 6 Marines (enemies), and 3sv4z, which features 3 Stalkers (allies) versus 4 Zealots (enemies). The win rates of the demonstrations in both cases are approximately 30%.

**How does PegMARL scale with joint demonstrations?** Figure 9 presents the learning curves of PegMARL alongside the DM² baseline in both the 5mv6m and 3sv4z configurations within the SMAC environment. When the joint demonstrations are sampled from **co-trained** policies, PegMARL achieves a comparable, and in some cases even higher, success rate compared to DM² in both tasks.

When the joint demonstrations are sampled from policies that are **non-co-trained**, the success rate of DM² exhibits a significant decline in both tasks. In the 5mv6m task, DM²'s success rate drops to nearly 0, indicating a failure to learn. In contrast, PegMARL maintains a similar level of performance compared to results from co-trained demonstrations, with only a slight decrease in convergence speed. These findings underscore the versatility and effectiveness of PegMARL in scenarios requiring collaboration among diverse agents, demonstrating its applicability and robustness across various multi-agent scenarios and demonstration types.

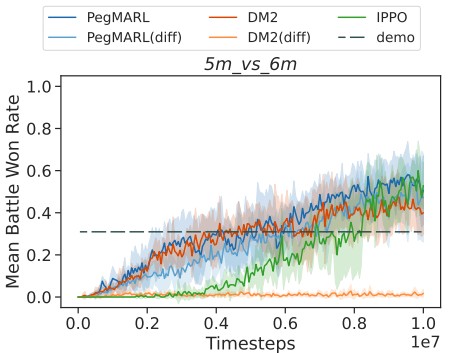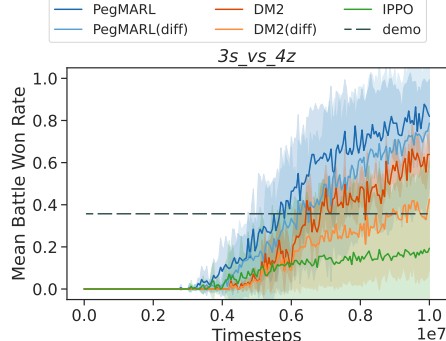

Figure 9: Learning curves of PegMARL versus DM$^2$ in two tasks under the **SMAC scenarios**. The suffix "diff" in the legend indicates that the joint demonstrations used are sampled from non-co-trained policies. Otherwise, the demonstrations are sampled from co-trained policies.

## 6 Conclusions

In this work, we introduce PegMARL, a novel approach for Multi-Agent Reinforcement Learning, which adopts personalized expert demonstrations as guidance and allows agents to learn to cooperate. Specifically, the algorithm utilizes two discriminators to dynamically reshape the reward function: one provides incentives to encourage the alignment between the policy behavior with provided demonstrations, and the other regulates incentives based on whether the behavior leads to the desired objective. We demonstrate PegMARL's effectiveness, scalability, and robustness in both discrete and continuous environments, where it outperforms state-of-the-art decentralized MARL algorithms, pure imitation learning, and reward-shaping techniques. We observe that PegMARL can achieve near-optimal policies even with suboptimal demonstrations. Furthermore, We showcase PegMARL's capability to leverage joint demonstrations and converge successfully, regardless of whether they are sampled from co-trained or non-co-trained policies.

**Applicability and Limitations:** PegMARL is most applicable when: a) The task allows for some degree of independent decision-making, allowing individual agent behaviors to be isolated and demonstrated, even if only partially. b) The environment provides feedback on cooperative behaviors, enabling agents to learn coordination beyond individual demonstrations. For example, in the door scenario, agents gain exploration guidance for door operation and navigation from personalized demonstrations. At the same time, environmental feedback on joint success drives them to learn coordination about timing and positioning (e.g., the green agent learning to make way for the red agent). Similarly, in the lava scenario, personalized demonstrations provide guidance for lava-avoiding paths. Still, environmental penalties for collisions and rewards for collective goal achievement encourage the emergence of cooperative navigation strategies.

However, PegMARL may face challenges in tasks requiring continuous, seamless collaboration among agents, such as cooperative object lifting and relocation by multiple agents. In these scenarios, where constant, integrated teamwork is crucial, the reliance on personalized demonstrations may not provide sufficient guidance for complex, ongoing coordination. Despite this, PegMARL's capability to utilize joint demonstrations from diverse sources, including non-co-trained policies, suggests the potential for addressing this limitation. It would be interesting future work to explore PegMARL's capabilities to handle more intricate and continuously collaborative tasks.

Another limitation of our work is the absence of formal convergence guarantees. While DM$^2$ establishes convergence for individual agents optimizing distribution matching objectives, our approach introduces additional complexity through the transition discriminator's dynamic adjustment of demonstration influence. The interplay between the dual discriminators and multi-agent dynamics poses challenges for theoretical analysis, particularly since the joint distribution cannot be directly recovered from individual agent demonstrations in our setting. While our extensive empirical results demonstrate robust convergence across diverse scenarios, establishing theoretical convergence guarantees remains an interesting direction for future work.

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

# A Environment Details

## A.1 Discrete Gridworld Environments

We use two gridworld environments with discrete state and action space: the lava scenario and the door scenario (Figure 5). The agents in the lava scenario are homogeneous because they have the same objective: navigating to their corresponding goal location. The door scenario has heterogeneous agents: the assistant agent (green) must open the door while the other agent (red) must reach the goal.

- **The lava scenario:** This environment has a 6-by-6 lava pond in the center (Figure 5a). We provide three variations of this scenario, each with varying numbers of agents and increasing levels of complexity. The main goal for the agents is to efficiently navigate to their assigned goal locations while avoiding stepping into the lava. An episode terminates when all agents reach their respective goals (succeed), or if any agents step into the lava or the maximum episode length is reached (fail).

- **The door scenario:** This environment is adapted from Franzmeyer et al. (2022) (see Figure 5b). In this scenario, the green agent must navigate to a designated green square and maintain its presence there to sustain the open state of the green door, thereby enabling the entry of a red agent into the right side room. An episode ends when the red agent reaches the red goal location (succeed) or the maximum episode length is reached (fail).

Each agent's local state space is its $\{x, y\}$ coordinates in the map. We concatenate all the local states together to form the global state and assume all agents have access to the global state, which has a dimension of $\mathbb{R}^{n \times 2}$ ($n$ is the agent number). The local action space includes five actions: left, right, up, down, and stay. A sparse reward is granted when an episode succeeds, while a small penalty will be subtracted according to the steps taken ($10 - \text{step\_count}/\text{max\_step}$). Agents will receive a penalty of $-1$ if they collide with each other.

## A.2 Cooperative Navigation

We modified the cooperative navigation task of multi-agent particle environment Lowe et al. (2017); Mordatch & Abbeel (2017) to evaluate the performance of our algorithm in a continuous environment. The modified environment consists of 2 agents, 2 goal landmarks and 1 wall between the agents and the goals. The agents need to navigate around the wall to occupy both landmarks. Each agent has a partial observation of the environment, including its position and velocity as well as the relative positions of other agents and landmarks. The action space is discrete, consisting of moving left, right, up, down, and staying in place. We sparsify their original reward as follows:

$$r = \sum_{a \in A} \min(0.3, \min_{g \in G} d(a, g)) + \sum_{g \in G} \min(0.3, \min_{a \in A} d(a, g)). \tag{11}$$

## A.3 The SMAC Environment

The StarCraft Multi-Agent Challenge (SMAC) (Samvelyan et al. (2019)) is an environment specifically designed for evaluating multi-agent reinforcement learning (MARL) algorithms. Built on the StarCraft II game, SMAC presents a variety of challenging scenarios where multiple agents must collaborate to control individual units in combat. Each agent has partial observation of its surroundings within a certain sight range. The action space is discrete, including move[direction], attack[enemy\_id], stop, and no-op. A global reward is given at each timestep based on the total damage dealt to the enemy units, with an additional bonus of 10 or 200 awarded for killing each enemy unit or for winning a combat respectively. We consider the same two tasks as used in DM2 (Wang et al. (2023)):

- **5m\_vs\_6m**: The allied team consists of 5 Marines and the enemy team controls 6 Marines.
- **3s\_vs\_4z**: The allied team controls 3 stalker units and faces 4 enemy Zealot units.

# B    Demonstration Details

## B.1    Personalized Demonstrations

We let each individual agent perform their designated task in the same environment map without the other agents' presence to collect personalized demonstrations. This can be implemented through two main methods: a) recording expert human demonstrations for each agent type, and b) training single-agent RL in simplified environments that focus on their specific tasks. In our experiments, we adopted the latter option. It takes about 2 hours to train a single-agent PPO for each personalized task.

**Discrete Gridworld Environments**: Figure 10 shows an example of personalized demonstrations for the lava scenario, and Figure 2 visualizes the personalized demonstrations for the door scenario. We summarize the details of the suboptimal demonstrations for the gridworld environment in Table 1, where the average episodic rewards are approximately 4.5, about half of their optimal counterparts.



Figure 10: An example of personalized demonstrations for the lava scenario (we did not visualize all the optimal paths). There is only one agent in the environment. The agent may take both the left and the right path toward the goal.

| Agent Id | $\mathcal{S}$ | $\mathcal{A}$ | Samples | Average Episodic Reward |
|----------|------|------|---------|-------------------------|
| 1 | $\mathbb{R}^2$ | 5 | 300 | 4.42 |
| 2 | $\mathbb{R}^2$ | 5 | 440 | 4.79 |
| 3 | $\mathbb{R}^2$ | 5 | 344 | 4.3 |
| 4 | $\mathbb{R}^2$ | 5 | 360 | 4.41 |

(a) The lava scenario

| Case | Agent | $\mathcal{S}$ | $\mathcal{A}$ | Samples | Average Episodic Reward |
|------|-------|------|------|---------|-------------------------|
| easy | red | $\mathbb{R}^2$ | 5 | 643 | 4.01 |
| easy | green | $\mathbb{R}^2$ | 5 | 612 | 4.39 |
| hard | red | $\mathbb{R}^2$ | 5 | 607 | 4.34 |
| hard | green | $\mathbb{R}^2$ | 5 | 593 | 4.36 |

(b) The door scenario

Table 1: The details of suboptimal demonstrations for the gridworld environment.

We provide visual representations of the policy and state occupancy measure corresponding to suboptimal demonstrations in Figure 11 for the lava scenarios and in Figure 12 for the door scenario. The red square symbolizes the agent's initial position in these visualizations, while the green square designates its respective goal location. Arrows within the figures denote available actions at each state, with arrow length indicating the probability associated with each action.

**Cooperative Navigation**: For the cooperative navigation task from the multi-agent particle environment (Lowe et al. (2017); Mordatch & Abbeel (2017)), the collaborative goal is for the two agents to cover both goals, regardless of which agent covers which goal. Therefore, we design the personalized task such that a single agent covers either of the two goals without the other agent's presence. An illustration of personalized demonstration for this task is shown in Figure 13.

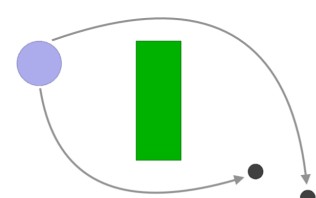

## B.2    Joint Demonstrations

**The SMAC Environment**: Regarding the joint demonstrations utilized in the SMAC environment (Samvelyan et al. (2019)), we adopted the demonstrations provided by the authors of DM2 (Wang et al. (2023)). For joint demonstrations sampled from co-trained policies, they are derived from jointly trained expert policies that achieve approximately a 30% win rate. For joint demonstrations sampled from non-co-trained policies, they are obtained from expert policies that were trained independently in separate teams but executed together in the same environment.

Figure 13: An illustration of personalized demonstrations for the cooperative navigation task. There is only one agent in the environment. The agent's objective is to reach one of the two goal locations.

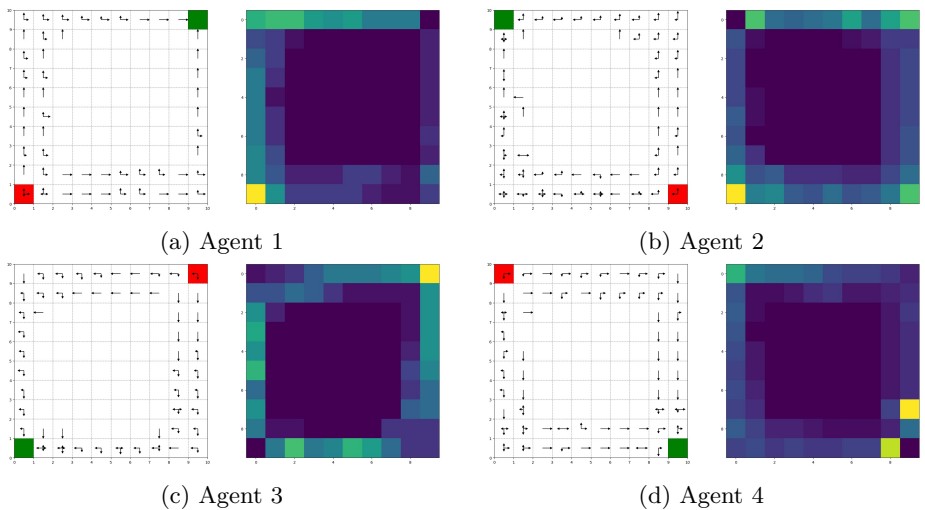

(a) Agent 1          (b) Agent 2

(c) Agent 3          (d) Agent 4

Figure 11: Agent policies and state occupancy measures estimated from the suboptimal demonstrations for the **lava scenario**.

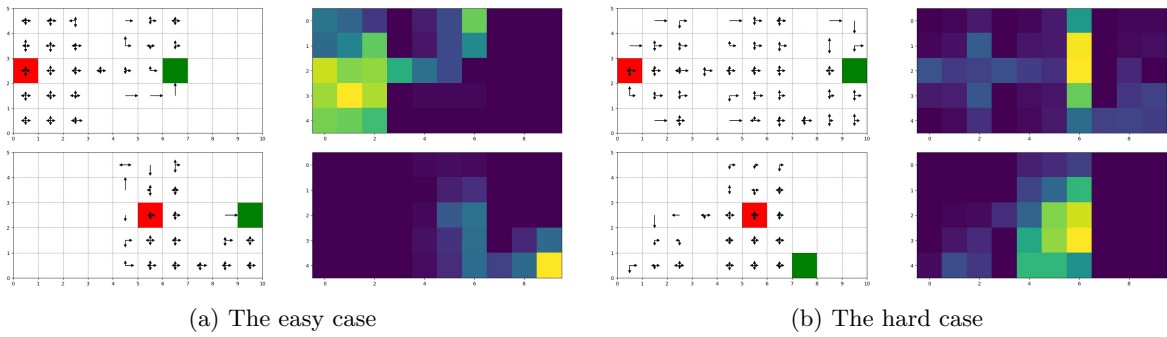

(a) The easy case          (b) The hard case

Figure 12: Agent policies and state occupancy measures estimated from the suboptimal demonstrations for the **door scenario**. The top row is for the red agent, and the bottom row is for the green agent.

## C    Network architecture and hyperparameters

### C.1    Training with Personalized Demonstrations

The algorithms are implemented based on MAPPO (Yu et al. (2021)), with each agent having separate policy and critic networks, discriminators, and optimizers. Both the policy and critic networks are two-layer MLPs with a hidden dimension of 64 and Tanh activation. The discriminators, including the personalized behavior discriminator and the personalized transition discriminator, are three-layer MLPs with a hidden dimension of 64 and Tanh activation. The personalized behavior discriminator takes $(s_i, a_i)$ as input, while the personalized transition discriminator takes $(s_i, a_i, s_i')$ as input.

In partially observable settings, let's assume that the agent's observation in the multi-agent MDP is denoted as $o_i \in \mathcal{O}_i$, while in the PerMDP, it is denoted as $o_i' \in \mathcal{O}_i'$ . Since the PerMDP involves a single agent and the multi-agent MDP involves multiple agents, the observation space for each agent cannot be the same in these two settings. For experiments on cooperative navigation, we use a heuristic function to convert $o_i$ by removing the dimensions that is not within the observation space $\mathcal{O}_i'$ before passing them to the discriminators.

All baseline algorithms, except ATA, use the same codebase we implemented. MAGAIL is trained with the personalized behavior discriminator but does not use the environmental reward. DM2 uses both the environmental reward and the personalized behavior discriminator. PegMARL utilizes the environmental reward, the personalized behavior discriminator, and the personalized transition discriminator. ATA's original implementation is adopted for comparison. We summarized the common hyperparameters used for PegMARL, DM2 and MAGAIL in Table 2. The *gail rew coef* in DM2 is set to be 0.02 for all the gridworld scenarios and 0.1 for the cooperative navigation task. The $\eta$ coefficient in PegMARL is 0.05 for all the gridworld scenarios and 0.2 for the cooperative navigation task.

| epochs | 4 |
|---|---|
| buffer size | 4096 |
| clip | 0.2 |
| lr | 0.0001 |

Table 2: MAPPO Hyperparamters

### C.2    Training with Joint Demonstrations

We implement PegMARL based on the original DM2's codebase to conduct experiments in the SMAC environment, augmenting it with our personalized transition discriminator. The hyperparameters for IPPO and GAIL were set to be identical to those used in DM2, as shown in Table 3 and 4. PegMARL adopted the same hyperparameters as DM2.

|  | 5v6 | 3sv4z |
|---|---|---|
| epochs | 10 | 15 |
| buffer size | 1024 | 1024 |
| gain | 0.01 | 0.01 |
| clip | 0.05 | 0.2 |

Table 3: DM2's IPPO Hyperparameters

|  | 5v6 | 3sv4z |
|---|---|---|
| gail rew coef | 0.3 | 0.05 |
| discr epochs | 120 | 120 |
| buffer size | 1024 | 1024 |
| batch size | 64 | 64 |
| n exp eps | 1000 | 1000 |

Table 4: DM2's GAIL Hyperparameters

### C.3    The influence of the weighting term $\eta$

In our experiments, we chose $\eta$ through a grid search to optimize performance. Interestingly, we found that PegMARL is relatively insensitive to $\eta$, which we attribute to the dynamic weighting provided by the personalized transition discriminator. This discriminator adaptively adjusts the influence of demonstration

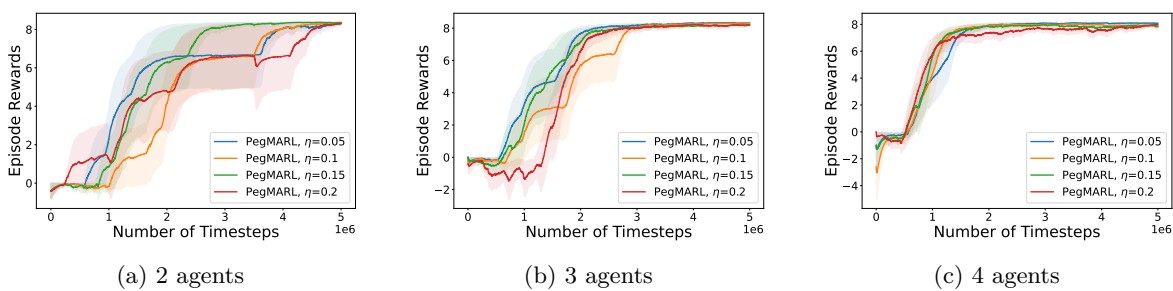

(a) 2 agents

(b) 3 agents

(c) 4 agents

Figure 14: Ablation study over the weighting term $\eta$ on the lava environment using optimal personalized demonstrations.

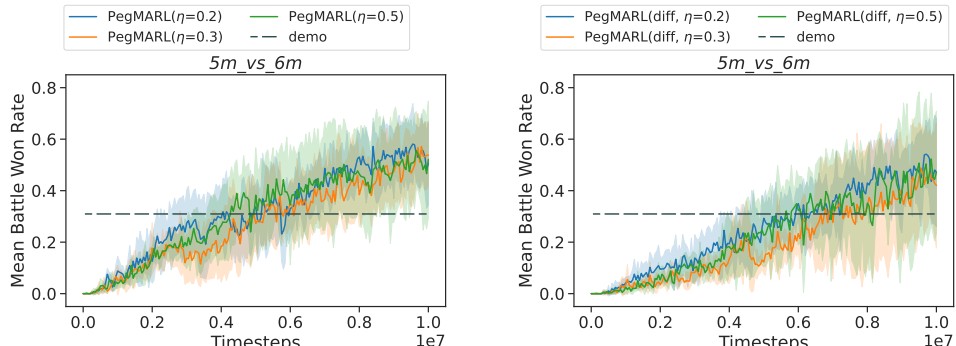

Figure 15: Ablation study over the weighting term $\eta$ on the 5m__vs__6m SMAC map. Left: results using joint demonstrations sampled from co-trained policies; Right: results using joint demonstrations sampled from non-co-trained policies.

alignment based on the likelihood of local state-action pairs leading to desired outcomes, thereby mitigating the impact of suboptimal $\eta$ values.

# D    Additional Visualizations

We additionally depict the state visitation frequencies of the joint policies learned by PegMARL with suboptimal demonstrations and MAPPO for both the lava (Figure 16) and the door scenario (Figure 17).

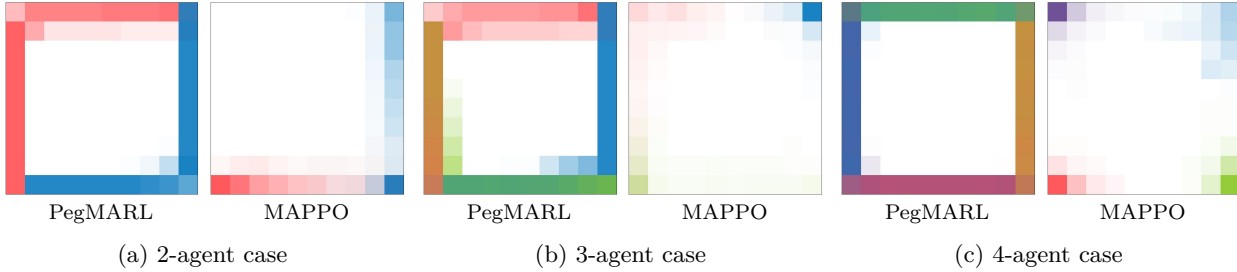

(a) 2-agent case

(b) 3-agent case

(c) 4-agent case

Figure 16: State visitation frequency of the joint policies learned by PegMARL (with suboptimal demonstrations) and MAPPO for the **lava scenario**. The darker color means a higher value. MAPPO failed to learn any meaningful policies in all three settings.

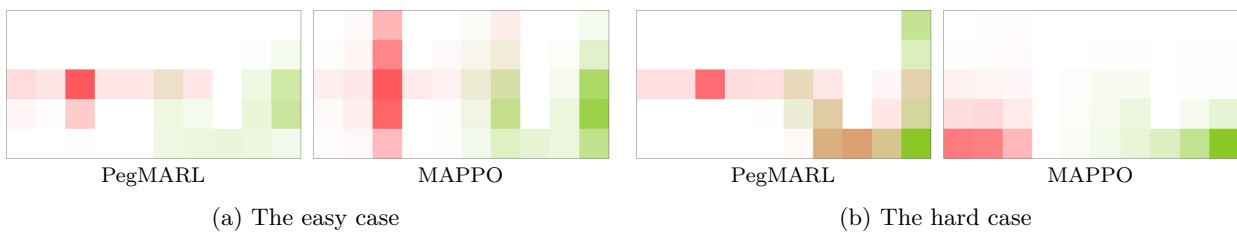

(a) The easy case

(b) The hard case

Figure 17: State visitation frequency of the joint policies learned by PegMARL (with suboptimal demonstrations) and MAPPO for the **door scenario**. The darker color means a higher value. MAPPO converges to a suboptimal policy in the easy case, while failing to learn in the hard case.

