# OpenReview forum: "Beyond Joint Demonstrations: Personalized Expert Guidance for Efficient Multi-Agent Reinforcement Learning"
_TMLR — Accepted by TMLR_

### Review · Reviewer_LZmN · 2024-07-24

**Summary Of Contributions:**

This paper studies the problem of using personalized expert demonstrations that pertain to single-agent behaviours, to improve cooperation in multi-agent cooperative environments. The objective is to use the available expert demonstrations to learn to cooperate using an algorithm that tries to align the behaviours of the agent to the behaviours of the demonstrations and improve the policy using reward shaping. The associated algorithm: PegMARL is tested on several cooperative multi-agent tasks including both discrete and continuous action spaces. PegMARL is demonstrated to outperform several state-of-the art MARL baselines in a series of multi-agent environments.

**Audience:**

Yes

**Broader Impact Concerns:**

N?A

**Claims And Evidence:**

Yes

**Requested Changes:**

Please address my comments in the Weaknesses above.

**Strengths And Weaknesses:**

Strengths:
The paper is well-motivated and mostly well-written. The experimental results are comprehensive. The figures in the paper effectively conveys the key points of the paper.


Weaknesses:

Overall I have a positive view of the paper. However, I think the paper still has some important limitations described as follows:

1. The distinction between homogeneous and heterogeneous agents is confusing throughout the paper. How exactly do the authors make this distinction (do heterogeneous agents have different state space, action space, and reward functions)? Further, why is this distinction not reflected in the personalized MDP setting described in Section 3?

2. I did not fully understand the Lava scenario. While the paper describes some increase in complexity and the number of agents, I am not sure I follow the argument here. All the three sub-figures in Figure 5(a) look
similar (why is one more complex than the other)? Also when testing for increasing numbers of agents in such a simple setting, more agents should have been considered (at-least tens of agents as opposed to just four agents). How is the expert guidance obtained in this setting?

3. The baselines and their implementations have not been clearly described. What are the major reasons for superior performances of PegMARL seen in Figure 6?

4. The experimental section does not clearly describe the expert guidance in any of the environments? How is expert guidance obtained in each environment? What is different in the case of StarCraft (why is it called
joint demonstrations)?

5. If the guidance is not accurate, why are these called expert guidances? For instance in StarCraft it is described that only 30% win rate is obtained by the demonstrations.

6. In the Introduction, the paper claims that PegMARL is able to obtain near-optimal policies even with suboptimal demonstrations. I think this has not been demonstrated anywhere in the paper. Can the authors clarify where exactly does PegMARL show this behaviour?


Minors:

1. Section 1: We prove the effectiveness -> We demonstrate the effectiveness (this is not a theoretical work!).
2. Section 5.1: both discrete gridworld -> discrete gridworld
3. Section 5.1: Ways for agents to interact to increase -> Unclear.
4. Section 6: alignment between policy behavior -> alignment between the policy behavior
5. Section 2: Qiu et al. suggests -> Qiu et al. suggest

---

> ### Author Response · Authors · 2024-09-02
> **Response to Reviewer LZmN (1)**
>
> Thank you for your thorough review and constructive feedback. We appreciate your recognition of our paper's motivation, writing quality, and experimental comprehensiveness. We are committed to providing detailed responses that will enhance the clarity and impact of our work. Below, we address each of your points:
>
> >1. The distinction between homogeneous and heterogeneous agents is confusing throughout the paper. How exactly do the authors make this distinction (do heterogeneous agents have different state space, action space, and reward functions)? Further, why is this distinction not reflected in the personalized MDP setting described in Section 3?
>
> 1. **Distinction between homogeneous and heterogeneous agents**: We apologize for any confusion. In our work, heterogeneous agents have distinct roles or capabilities, potentially leading to different state/action spaces or reward functions. For simplicity, Section 3 presents the PerMDP definition for Individual agents which may or may not be heterogeneous, though it can be extended to agent types in heterogeneous settings. We have clarified this point more explicitly in the revised paper in Section 3.
>
> >2. I did not fully understand the Lava scenario. While the paper describes some increase in complexity and the number of agents, I am not sure I follow the argument here. All the three sub-figures in Figure 5(a) look similar (why is one more complex than the other)? Also when testing for increasing numbers of agents in such a simple setting, more agents should have been considered (at-least tens of agents as opposed to just four agents). How is the expert guidance obtained in this setting?
>
> 2. **Lava Scenario**: We apologize for the lack of clarity. The scenario's complexity increases with the number of agents due to the exponential growth in possible interactions and joint action space. Coordinating collision-free paths becomes more challenging as agents are added. Moreover, the risk of mission failure increases since any agent entering lava terminates the episode for all. We focused on up to four agents to demonstrate scalability. Each agent's task remains consistent (navigating diagonally), so complexity increase stems solely from agent interactions. For personalized demonstrations, we trained single-agent PPO in simplified environments with only one agent present (as illustrated in Figure 10). We’ve provided a more detailed explanation in the Experiments section of the revised paper.
>
> >3. The baselines and their implementations have not been clearly described. What are the major reasons for superior performances of PegMARL seen in Figure 6?
>
> 3. **Baselines and PegMARL's superior performance**: We’ve provided a more detailed description of the baselines and their implementations in the revised paper (in Experiments section 5.1 and Appendix C). PegMARL's superior performance stems from its dual discriminator approach, particularly the personalized transition discriminator. This allows for selective learning from demonstrations, effectively filtering out less desirable actions while retaining beneficial ones. In contrast, other methods may more directly imitate demonstrations, potentially replicating suboptimal actions.
>
> >4. The experimental section does not clearly describe the expert guidance in any of the environments? How is expert guidance obtained in each environment? What is different in the case of StarCraft (why is it called joint demonstrations)?
>
> 4. **Expert guidance in different environments**: We have added a clear description of how expert guidance was obtained in the Experiments section and Appendix B. For gridworld and cooperative navigation, we used single-agent PPO trained on simplified versions of the tasks. For StarCraft, we used joint demonstrations (trajectories from all agents simultaneously) sampled from co-trained and non-co-trained IPPO policies to compare with DM^2, which requires such demonstrations.
>
> >5. If the guidance is not accurate, why are these called expert guidances? For instance in StarCraft it is described that only 30% win rate is obtained by the demonstrations.
>
> 5. **Accuracy of expert guidance**: Thanks for this question. The term "expert guidance" in our context refers to demonstrations that provide meaningful, above-random performance in complex environments, even if not optimal. In StarCraft, achieving a 30% win rate is non-trivial given the game's complexity and the performance of the built-in AI. These demonstrations, while suboptimal, contain valuable information about effective strategies and behaviors. We have added this clarification as a footnote to the Introduction section of the paper.

---

> > ### Author Response · Authors · 2024-09-02
> > **Response to Reviewer LZmN (2)**
> >
> > >6. In the Introduction, the paper claims that PegMARL is able to obtain near-optimal policies even with suboptimal demonstrations. I think this has not been demonstrated anywhere in the paper. Can the authors clarify where exactly does PegMARL show this behaviour?
> >
> > 6. **Near-optimal policies with suboptimal demonstrations**: Thank you for highlighting this. In our gridworld experiments, PegMARL achieved oracle performance with suboptimal demonstrations, whose episodic returns are about half of their optimal counterparts. For SMAC experiments, in the 3s_vs_4z scenario, PegMARL reached approximately 80% success rate, significantly outperforming DM^2, which achieved only 40-60%. These results demonstrate PegMARL's ability to learn near-optimal policies from suboptimal demonstrations across different environments.
> >
> > >Minors:
> > >1. Section 1: We prove the effectiveness -> We demonstrate the effectiveness (this is not a theoretical work!).
> > >2. Section 5.1: both discrete gridworld -> discrete gridworld
> > >3. Section 5.1: Ways for agents to interact to increase -> Unclear.
> > >4. Section 6: alignment between policy behavior -> alignment between the policy behavior
> > >5. Section 2: Qiu et al. suggests -> Qiu et al. suggest
> >
> > **Minor corrections**: We thank you for these detailed suggestions and have implemented all of them in our revised manuscript.
> >
> > We believe these changes will address your concerns and strengthen the paper. Thank you again for your valuable feedback.

---

### Review · Reviewer_C3De · 2024-07-25

**Summary Of Contributions:**

The paper proposes PegMARL, an approach that leverages personalized (of individual agents) expert demonstrations for cooperative multi-agent reinforcement learning. The approach makes use of reward shaping, using two discriminators that aim at aligning each agent's policy with the demonstration. The approach also includes a parameter that allows to tune how much of the objective is the alignment an how much of it is the collective reward. Finally, the approach is validated in a couple of discrete and continuous environments and shown to outperform the baselines.

**Audience:**

Yes

**Broader Impact Concerns:**

I do not expect particular concerns.

**Claims And Evidence:**

No

**Requested Changes:**

I group the requested changes in two:
- first, the paper should include a discussion on how to obtain expert demonstrations, and in which types of scenarios obtaining them should be easy and not. In relation to this point, in the case of the experiments on this paper, it is important to know how much computation was needed to obtain the personalized demonstrations: MAPPO, for instance, that does not have that added computational time, so the comparison on efficiency may be unfair.
- second, it is important to have a discussion on the choice of $\eta$, as well as experimental validation for the robustness of the method to the parameter. Not exclusively, the authors should explain how their parameter $\eta$ was chosen. It would also be important to see a figure with a metric of performance (for instance the area under curve of the performance and the performance at the end of training) as a function of different values of $\eta$. Finally, how were the hyperparameters of the baselines chosen, for instance for MA2?

The requested changes are important and should be addressed by the authors in their rebuttal, as well as integrated in the document if the authors agree on their sense.

**Strengths And Weaknesses:**

The paper is very clear, the approach is well explained and it is also sound. Even though the experiments are not many, nor the tasks seem particularly difficult, there appears to be clear benefit over the baselines.

On the downside:
- the paper lacks a contextualization of in which scenarios personalized demonstrations can be obtained, and how to obtain them. The availability of personalized demonstrations significantly limits the approach. In multi-agent settings, the co-ordinated policy may frequently be completely unknown.
- there is no discussion or validation on the choice of the parameter $\eta$, which should be quite important for the results, and problem-dependent: if the demonstrations are very good and the global task is individually decomposable in personalized tasks, possibly $\eta$ should be higher. It would important to have experimental validation on the sensitivity of the algorithm to the parameter.

---

> ### Author Response · Authors · 2024-09-02
> **Response to Reviewer C3De (1)**
>
> Thank you for your thoughtful comments and suggestions. We appreciate your recognition of the clarity and soundness of our approach. We are committed to addressing your concerns and requested changes to further enhance the quality of our work. Below, we provide detailed responses to each of your points.
>
> >the paper lacks a contextualization of in which scenarios personalized demonstrations can be obtained, and how to obtain them. The availability of personalized demonstrations significantly limits the approach. In multi-agent settings, the coordinated policy may frequently be completely unknown.
>
> > first, the paper should include a discussion on how to obtain expert demonstrations, and in which types of scenarios obtaining them should be easy and not. In relation to this point, in the case of the experiments on this paper, it is important to know how much computation was needed to obtain the personalized demonstrations: MAPPO, for instance, that does not have that added computational time, so the comparison on efficiency may be unfair.
>
>
> 1. **Contextualization of using personalized demonstrations**: We agree that this is an important point to clarify. Personalized demonstrations are applicable in scenarios where individual agent behaviors can be isolated and recorded separately, even if only partially. These demonstrations, combined with environmental feedback, allow agents to learn both individual skills and cooperative behaviors. For instance, in our door scenario (Figure 1 and 5b), we can isolate and demonstrate individual tasks like navigation and door operation, while agents learn cooperation through environmental feedback. In the lava scenario (Figure 5a), demonstrations teach basic navigation without stepping into lava, while collision avoidance is learned from environmental rewards.
>
> 	Personalized demonstrations are particularly suitable when: a) heterogeneous teams require different demonstrations for different agent types, and b) existing single-agent expert policies are available but need to be adapted for multi-agent settings, c) environments have sparse rewards, where most existing methods struggle. Methods for obtaining personalized demonstrations include: a) recording expert human demonstrations for each agent type, and b) training single-agent RL in simplified environments that focus on their specific tasks. In our experiments, we adopted the latter option.
>
> 	While computational costs vary with method and task complexity, personalized demonstrations are generally much more cost-effective than collecting joint demonstrations. For instance, in our gridworld environment, training single-agent PPO for each personalized task takes about 2 hours to achieve optimal performance. In contrast, MAPPO struggles to solve the problem when attempting to collect joint demonstrations. We have added a detailed discussion on this in Appendix B.1 of our revised paper.
>
> 	Notably, PegMARL can also utilize joint demonstrations, including those from non-co-trained policies (see SMAC results in Figure 9). This capability is particularly valuable in real-world scenarios where high-quality demonstrations are scarce and often come from multiple, potentially incompatible sources. By accommodating these diverse and potentially conflicting demonstrations, PegMARL can leverage a wider range of available data, offering greater flexibility and robustness across various multi-agent applications.

---

> > ### Author Response · Authors · 2024-09-02
> > **Response to Reviewer C3De (2)**
> >
> > > there is no discussion or validation on the choice of the parameter 𝜂, which should be quite important for the results, and problem-dependent: if the demonstrations are very good and the global task is individually decomposable in personalized tasks, possibly 𝜂 should be higher. It would important to have experimental validation on the sensitivity of the algorithm to the parameter.
> >
> > > second, it is important to have a discussion on the choice of 𝜂, as well as experimental validation for the robustness of the method to the parameter. Not exclusively, the authors should explain how their parameter 𝜂 was chosen. It would also be important to see a figure with a metric of performance (for instance the area under curve of the performance and the performance at the end of training) as a function of different values of 𝜂. Finally, how were the hyperparameters of the baselines chosen, for instance for MA2?
> >
> >
> > 2. **Choice and sensitivity of η**: We acknowledge the importance of η in balancing demonstration alignment and collective reward. In our experiments, we chose η through a grid search to optimize performance. Interestingly, we found that PegMARL is relatively insensitive to η, which we attribute to the dynamic weighting provided by the personalized transition discriminator. This discriminator adaptively adjusts the influence of demonstration alignment based on the likelihood of local state-action pairs leading to desired outcomes, thereby mitigating the impact of suboptimal η values. We have included an ablation study in Appendix C.3 of the revised paper.
> >
> > 3. **Hyperparameter selection for baselines**: For fairness, we used recommended hyperparameters from the original papers for baselines like DM^2. A comprehensive description of hyperparameter selection for all baselines is provided in Appendix C.

---

### Review · Reviewer_ZE3i · 2024-08-19

**Summary Of Contributions:**

This paper studies a new multi-agent RL (MARL) problem – personalized expert-guided MARL, where the demonstrations are collected for each individual type of experts. By adapting the DM^2 method to this setting, the authors propose PegMARL, which learns the augmented reward for each agent via a GAIL-like subroutine. To handle the restricted domain, this paper proposes to approximate the indicator function by a second discriminator. The proposed PegMARL is evaluated in multiple MARL domains, including both discrete-action environments (like minigrid) and the continuous-action environment (like SMAC) against multiple decentralized MARL benchmark methods.

**Audience:**

Yes

**Claims And Evidence:**

Yes

**Requested Changes:**

Please see the weaknesses above.

Some additional detailed questions / comments:
- Eq. (6): How is the $\epsilon$ determined? This parameter does not appear anywhere in the formulation.
- What’s the difference between PegMARL and the modified DM^2 with personalized demonstrations in Section 5?

**Strengths And Weaknesses:**

Strengths

- PegMARL is an interesting new MARL problem that could be of interest to the RL community.
- The motivating examples are helpful in illustrating the main intuition of the paper.
- The experimental results are quite strong.

Weaknesses

- My main concern is on the feasibility of using only personalized demonstrations in MARL. Specifically, personalized demonstrations can be viewed as the marginals. However, in general, we know that the joint distribution cannot be recovered solely from the marginal distributions of each random variable, unless there is some additional condition, e.g., independence. Therefore, the problem appears somewhat ill-posed, and this limitation has not been well explained. It would be very helpful to describe to what extent PegMARL can work.

- Moreover, the resulting algorithm (cf. Algorithm 1) appears to be a direct extension of distribution matching in DM^2 to the PegMARL. However, given the first point mentioned above, the convergence property under joint demonstrations (i.e., convergence of independent GAIL learners in the DM^2 paper) no longer holds here. Intuitively, the algorithm shall have something fundamentally different to address the issue.

- The comparison in SMAC could be further explained, and there are several missing pieces: 1) How is PegMARL adapted to the setting of joint demonstrations? 2) Why can PegMARL outperform DM^2 with co-trained policies (given that DM^2 is designed specifically for this setting)? 3) The performance of DM^2 appears somewhat weaker than those reported in the DM^2 paper. Some explanation about this would be helpful.

---

> ### Author Response · Authors · 2024-09-02
> **Response to Reviewer ZE3i (1)**
>
> Thank you for your thorough review and insightful feedback. We are grateful for your recognition of our work's strong motivation, comprehensive experimental results, and potential impact on the RL community. We appreciate the opportunity to address these issues and are committed to providing a detailed response that will strengthen the paper's clarity and contribution to the field.
>
> > My main concern is on the feasibility of using only personalized demonstrations in MARL. Specifically, personalized demonstrations can be viewed as the marginals. However, in general, we know that the joint distribution cannot be recovered solely from the marginal distributions of each random variable, unless there is some additional condition, e.g., independence. Therefore, the problem appears somewhat ill-posed, and this limitation has not been well explained. It would be very helpful to describe to what extent PegMARL can work.
>
> 1. **Feasibility of using only personalized demonstrations**: We fully agree that recovering the full joint distribution from marginals alone is generally not possible. This is exactly why in our approach we do not aim to fully reconstruct the joint distribution. PegMARL is not based on pure imitation from the personalized demonstrations. Instead, it uses personalized demonstrations as an exploration guide for individual agents while allowing agents to learn cooperation through environmental interactions and receiving rewards from the environment. This is greatly helpful when the environment rewards are sparse which is indeed the case in our work.
>
> 	PegMARL is most applicable when: a) The task allows for some degree of independent decision-making, allowing individual agent behaviors to be isolated and demonstrated, even if only partially. b) The environment provides feedback on cooperative behaviors, enabling agents to learn coordination beyond individual demonstrations. For example, in the door scenario (Figure 1 and 5b), agents gain exploration guidance for door operation and navigation from personalized demonstrations. At the same time, environmental feedback on joint success drives them to learn coordination about timing and positioning (e.g., the green agent learning to make way for the red agent). Similarly, in the lava scenario (Figure 5a), personalized demonstrations provide guidance for lava-avoiding paths. Still, environmental penalties for collisions and rewards for collective goal achievement encourage the emergence of cooperative navigation strategies. These examples illustrate how PegMARL utilizes personalized demonstrations to guide basic skills exploration while facilitating the development of cooperative behaviors through environmental feedback. We have added a detailed discussion on these conditions and limitations in the conclusion section of the revised paper.
>
> > Moreover, the resulting algorithm (cf. Algorithm 1) appears to be a direct extension of distribution matching in DM^2 to the PegMARL. However, given the first point mentioned above, the convergence property under joint demonstrations (i.e., convergence of independent GAIL learners in the DM^2 paper) no longer holds here. Intuitively, the algorithm shall have something fundamentally different to address the issue.
>
> 2. **Algorithm distinctiveness and convergence properties**: Our algorithm differs from DM^2 in two key aspects:
>
> 	(a) The introduction of a personalized transition discriminator, which aligns individual actions with desired state transitions, thereby addressing the challenges of using personalized demonstrations in a multi-agent setting. This component is crucial in addressing the aforementioned issue that joint distribution cannot be recovered solely from marginals. It provides a mechanism to evaluate the likelihood of local state-action pairs leading to desired outcomes in the context of the multi-agent environment, effectively bridging the gap between individual demonstrations and joint behavior.
>
> 	 (b) A stronger emphasis on learning cooperative behaviors through environmental interactions, rather than primarily relying on demonstrations. This approach allows for the emergence of coordinated strategies that may not be explicitly present in the personalized demonstrations. The transition discriminator dynamically balances between imitation and environmental rewards, favoring imitation when transitions align with demonstrations and vice versa.
>
> 	These key differences enable PegMARL to effectively utilize a diverse range of demonstrations, including personalized, jointly coordinated, and crucially, mixed-source demonstrations. Unlike DM^2, which is limited to demonstrations from co-trained policies, PegMARL can handle potentially incompatible behaviors from mixed-source policies.  (to be continued)

---

> > ### Author Response · Authors · 2024-09-02
> > **Response to Reviewer ZE3i (2)**
> >
> > This capability is particularly valuable in real-world scenarios where demonstrations may come from multiple sources or when agents trained independently must interact. By accommodating such diverse and potentially conflicting demonstrations, PegMARL offers greater flexibility and robustness across various multi-agent applications.
> >
> > While the convergence properties for joint demonstrations don't directly apply to PegMARL, our empirical results (Figures 6-9) consistently show robust convergence across various scenarios. This would be interesting follow-up work to analyze the theoretical convergence properties of PegMARL and learning from personalized demonstrations more broadly.
> >
> > >The comparison in SMAC could be further explained, and there are several missing pieces: 1) How is PegMARL adapted to the setting of joint demonstrations? 2) Why can PegMARL outperform DM^2 with co-trained policies (given that DM^2 is designed specifically for this setting)? 3) The performance of DM^2 appears somewhat weaker than those reported in the DM^2 paper. Some explanation about this would be helpful.
> >
> >
> > 3. **SMAC comparisons**: Thank you for highlighting these important points. We provide the answers here and also addressed them in the revised paper:
> >
> > 	(1) ***PegMARL's adaptation to joint demonstrations***: In our presented algorithm, the local state s_i is used as the discriminators’ input. However, in practice, agents often have access to observations (o_i) rather than the states. When using personalized demonstrations, we process the observations from joint rollouts in the joint MDP by removing dimensions not observable in the personalized MDP. This step is unnecessary for the case using joint demonstrations as both demonstrations and environment rollouts come from the joint MDP.
> >
> > 	(2) ***Performance vs DM^2 with co-trained policies***: PegMARL's superior performance can be attributed to its dual discriminator approach, which provides more nuanced guidance. Given that the joint demonstrations have only a 30-40% success rate, they likely contain suboptimal behaviors. PegMARL's transition discriminator allows for selective learning, effectively filtering out less desirable actions while retaining beneficial ones. In contrast, DM^2 more directly imitates the demonstrations, potentially replicating suboptimal actions. This key difference enables PegMARL to extract more value from imperfect demonstrations, leading to improved performance over DM^2, especially when high-quality demonstrations are scarce.
> >
> > 	(3)***DM^2’s performance discrepancy***: We've verified that our DM^2 results are in general consistent with the original paper. We obtained the joint demonstrations used in the DM^2 paper's ablation experiments (Figure 3) from the authors and used the lower success rate demonstrations in our experiments. We adopted DM^2's original implementation and parameters for accurate replication. A direct comparison between our Figure 9 and the DM^2 paper's Figure 3 shows comparable performance levels, with minor variations. For instance, in the 5m_vs_6m map, using demonstrations with a ~30% success rate, DM^2 and DM^2(diff) achieved ~45% and ~0% success rates respectively at the end of training in our experiments. This is similar to the original paper's results for dm2(concur., joint) and (concur., indep), which achieved ~50% and ~0% respectively.
> >
> >
> > >Some additional detailed questions / comments:
> > > - Eq. (6): How is the 𝜖 determined? This parameter does not appear anywhere in the formulation.
> > > - What’s the difference between PegMARL and the modified DM^2 with personalized demonstrations in Section 5?
> >
> >
> > 4. **Additional questions**:
> >
> > 	(a) **Eq. (6) 𝜖:** This equation defines a restricted domain using JS divergence to ensure occupancy matching occurs only for local state-action pairs leading to desired outcomes. While 𝜖 theoretically controls transition mismatch tolerance, directly verifying this constraint is impractical. Instead, we introduce the personalized transition discriminator as a practical approximation. This discriminator learns to assess the likelihood of local state-action pairs leading to desired next states, effectively serving as a learned proxy for the 𝜖-constrained domain in Equation 6.
> >
> > 	(b) **PegMARL vs modified DM^2 with personalized demonstrations**: The key distinction between PegMARL and the modified DM^2 lies in PegMARL's incorporation of a personalized transition discriminator. This component, as detailed in our previous response to question 2, plays a crucial role in aligning individual actions with desired state transitions and balancing imitation with environmental learning.

---

### Decision · Action_Editor_mpYi · 2024-10-15

**Recommendation:** Accept with minor revision

**Comment:**

- One common criticism in the first round of reviews was that it was not clear how to obtain the expert demonstrations. Authors seemed to have incorporated explanations on this issue in the manuscript, which is good.

- One concern from the reviewers that does not seem to be addressed is about the following claim that appears in the Abstract, Introduction (contribution number (2)) and Conclusion. The text claims:

> "The algorithm obtains near-optimal policies even when provided with suboptimal demonstrations."

However it is not clear what this refers to. Can you please add justifications/explanations for this claim and what it refers to in the results? Conclusion states that this above claim is observed but it is hard to find a clear explanation in Section 5. Please address this in the revision and clearly explain which experiments this claim refer to.

- Another concern is about the lack of convergence guarantees. This is already acknowledged in your rebuttal, can you please incorporate this discussion in your main text and describe difficulties on getting convergence guarantees compared to previous work? This is an empirical work and the reviewing team agrees that the empirical contribution is sufficient but it is good to still have explanations about theory to motivate future work.

- Please explain, so your readers understand better, how Eq. 9 and Eq. 10 are derived and how they are computed in practice. Looking at the reference Kang et al. (2018), one can see where these equations come from but it is better to make your paper self-contained.

- One reviewer pointed out the similarities between the proposed algorithm and DM^2 from Wang et al. (2023). The authors acknowledged this and explained the differences in their rebuttal. I am not sure if this discussion is incorporated in the text. In particular, please include this discussion when you introduce Algorithm 1 to highlight the similarities and differences compared to DM^2.

- Two non-technical suggestions. 1. Please change phrases such as "However, blindly imitating, ..." which appears in Section 4.1. You may say "However, only imitating, ..." 2. Typo in the title of Appendix D: "Addtional"

**Audience:**

The reviewing team agrees that the findings of this paper are interesting to some individuals in TMLR's audience.

**Claims And Evidence:**

The reviewing team agreed that the paper mostly satisfies this criteria. The reason for "mostly" is that there is a need for a minor revision to fully address some concerns in the reviews. I think that these are minor changes which is the reason behind the decision. These changes include either rephrasing a claim in the abstract and introduction or adding more explanations in the text to clarify this claim. I will highlight the details of the minor revision in the "Comments" section.

---

> ### Author Response · Authors · 2024-11-20
> **Response to Action Editor mpYi**
>
> We thank the action editor for their thoughtful feedback. We have made the following revisions to address their comments:
>
> 1. We have revised performance-related claims to be more empirically focused, changing phrases like "near-optimal policies" to more precise language about empirical performance.
> 2. We have added a discussion of theoretical considerations to the conclusion section, acknowledging current limitations while highlighting the unique challenges posed by using personalized demonstrations and our dual-discriminator approach compared to existing work.
> 3. We have included a detailed derivation of Equation (7) in Appendix E, which forms the theoretical foundation for the practical optimization objectives in Equations (9) and (10).
> 4. We have added explicit comparison with DM^2 in Section 4.2, highlighting key differences in our approach especially regarding the personalized transition discriminator and handling of personalized and non-co-trained joint demonstrations.
> 5. We have made editorial improvements, including language refinements and typo corrections.
>
> We hope these revisions have improved the clarity and technical precision of the paper while better contextualizing our contributions within the field.